# Magnification of Iris through Clear Acrylic Resin in Ocular Prosthesis

**DOI:** 10.3390/jfb13010029

**Published:** 2022-03-11

**Authors:** Dinesh Rokaya, Jidapa Kritsana, Pokpong Amornvit, Nagendra Dhakal, Zohaib Khurshid, Muhammad Sohail Zafar, Preamjit Saonanon

**Affiliations:** 1Department of Clinical Dentistry, Walailak University International College of Dentistry, Walailak University, Bangkok 10400, Thailand; 2Oculoplastic and Reconstructive Surgery Unit, Department of Ophthalmology, Faculty of Medicine, King Chulalongkorn Memorial Hospital, Chulalongkorn University, Bangkok 10330, Thailand; jidapa.kri@gmail.com (J.K.); psaonanon@gmail.com (P.S.); 3PPFACEDESIGN Center, Sathorn, Bangkok 10120, Thailand; pokpong_am@yahoo.com; 4Department of Physics, University of Central Florida, Orlando, FL 32816, USA; nagendra.dhakal@ucf.edu; 5Department of Prosthodontics and Implantology, College of Dentistry, King Faisal University, Al-Hofuf 31982, Al Ahsa, Saudi Arabia; zsultan@kfu.edu.sa; 6Department of Restorative Dentistry, College of Dentistry, Taibah University, Al Madinah 41311, Al Munawwarah, Saudi Arabia; drsohail_78@hotmail.com; 7Department of Dental Materials, Islamic International Dental College, Riphah International University, Islamabad 44000, Pakistan

**Keywords:** ocular prosthesis, eye prosthesis, iris, magnification, clear acrylic, acrylic resin, polymethyl methacrylate

## Abstract

The study on the magnification of the clear acrylic resin in prosthetic dentistry is important but lacking. Hence, this research aimed to investigate the magnification of the iris in the ocular prosthesis with various thicknesses of clear heat cure acrylic resin. A total of 60 ocular prostheses were divided into six groups with various thicknesses of clear heat cure acrylic resin over the iris; T0 (control): no acrylic resin, T1: 2, T2: 4, T3: 6, T4: 8, and T5: 10 mm of clear acrylic resin over the iris. Each ocular prosthesis was manufactured from white acrylic, with a 10.250 mm iris. Then, the clear heat cure resin was placed over the iris, cured, then polished. The final diameter of the iris was measured with a vernier caliper. The sizes of the iris were compared among various groups using one-way ANOVA, and a significant level was chosen at *p* value = 0.01. The mean sizes of iris were T0 = 10.25, T1 = 10.92, T2 = 11.26, T3 = 11.91, T4 = 12.75, and T5 = 13.43 mm. The size of the iris was significantly different among the group (*p* < 0.0001). The magnification of the iris for different groups was 1.06 for T1, 1.10 for T2, 1.16 for T3, 1.24 for T4, and 1.31 for T5. The magnification of the iris increased as the thickness of clear heat cure acrylic resin over the iris is increased on the ocular prosthesis.

## 1. Introduction

Anophthalmic sockets or loss of an eyeball can be caused by various reasons, mainly due to trauma (46%) and malignant tumor (44%), followed by nonspecific (8%) and congenital (2%) [1,2]. Among malignant tumors, retinoblastoma and malignant melanoma were the most common pathological diagnoses [3,4]. The prosthetic rehabilitation of an ophthalmic socket is important, as an eye defect undeniably impacts patient self-esteem and causes esthetic and psychological problems [5,6]. An ocular prosthesis, or fake eye (Figure 1), generally fabricated from polymethyl methacrylate (PMMA) or acrylic resin, replaces the missing organ and provides esthetics and psychological comfort to the patient. The ocular prosthesis of appropriate contour, size, and color bars patients from being handicapped and allows them to pursue their normal lives [7,8].

The ocular prosthesis can be prefabricated or custom-made [9]. The latter provides a better fit to the eye socket, better cosmetic results, and more comfort to the patients. The custom-made prosthesis can be fabricated by conventional techniques or assisted with digital technologies [10,11]. In the conventional technique, following the impression of the ocular defect, the base shell is fabricated following curing, and staining is done manually to match the patient’s natural eye [11]. In digitally-assisted techniques, the base shell can be 3D printed, and staining can be assisted and printed with digital photography [12,13]. Following staining, in both fabrication techniques, the addition and curing of a thin superficial layer of acrylic resin are required to resemble the human cornea [9,10].

The common problems of fitting the ocular prosthesis are in the proper shape or size of the prosthesis, as well as the difficulty in matching color, eyelid aperture, and iris size to the normal eye. Rasmussen [14] found that the most frequent complications associated with ocular prostheses are secretion, lagophthalmos, enophthalmos, rotating prosthesis, prosthesis falling out, and exophthalmos. Anophthalmic patients are likely to experience problems with their prosthetic eye, and a survey on patients’ concerns and satisfaction with custom ocular prostheses found that the most common prosthesis-related concerns were reduction of eye motility, eye discharge, and difference in the size of the prosthetic eye, relative to the other eye [15].

The larger-than-expected iris size is usually caused by the magnification of the iris during the fabrication of a new ocular prosthesis and relining of the old ocular prosthesis which results in an unesthetic prosthesis (Figure 2). The clear acrylic over the iris acts as a lens and causes the magnification of the iris. However, the research on the magnification of the clear acrylic resin is still lacking. Hence, this study aimed to investigate the magnification of the iris in the ocular prosthesis with various thicknesses of clear heat cure acrylic resin. Ocular prostheses were equally divided into six groups with various thicknesses of clear heat cure acrylic resin (polymethyl methacrylate, PMMA) over the iris; no acrylic resin over the iris (control); and 2, 4, 6, 8, and 10 mm thickness of clear acrylic resin over the iris. The sizes of the iris among groups were compared. It was found that the magnification of the iris increases significantly as the thickness of clear heat cure acrylic resin over the iris is increased on the ocular prosthesis.

## 2. Materials and Methods

### 2.1. Materials and Instruments

The acrylic resin (white and clear polymethyl methacrylate) and coloring materials for the fabrication of ocular prosthesis were bought from the sclera polymer, Factor II Inc., Lakeside, AZ, USA. A vernier caliper (Mitutoyo Co., Kanagawa, Japan) was used to measure the iris size (diameter).

### 2.2. Fabrication of Ocular Prostheses

A total of 60 ocular prostheses were equally divided into six groups: T0, T1, T2, T3, T4, and T5, consisting of the ocular prosthesis with various thicknesses of clear heat cure acrylic resin over the iris, as shown in Table 1.

All subjects gave their informed consent for inclusion before they participated in the study. The study was conducted in accordance with the Declaration of Helsinki. Informed consent was obtained from all subjects involved in the study.

At first, ocular molds were prepared with dental stones. Then, the white acrylic resin was packed and cured, according to the manufacturer’s recommendation. Following curing, the ocular prostheses were trimmed and polished. The center of the ocular prosthesis was marked in each ocular prosthesis. The specific depth portions of acrylic at the center of the white acrylic prosthesis were removed: none for T0, 2 mm for T1, 4 mm for T2, 6 mm for T3, 8 mm for T4, and 10 mm for the T5 group. Then, from a black A4-sized paper, a round iris, with a mean diameter of 10.25 mm (initial size of the iris), was cut and attached to each ocular prosthesis using glue (Figure 3 and Figure 4). The iris is covered with the monopoly (10 parts of monomer: 1 part of polymer) and dried.

After that, the clear heat cure resin was mixed and placed over the iris (at the dough stage) and allowed to cure, according to the manufacturer’s recommendation. Following curing, the ocular prostheses were polished.

### 2.3. Measurement of the Final Measured Diameter of the Iris in Ocular Prostheses

The final measured diameter of the iris of each ocular prosthesis was measured with a vernier caliper. Each measurement was done three times by one investigator, and the mean was recorded.

### 2.4. Calculation of the Magnification

The magnification of the iris in each ocular prosthesis was calculated by the following formula (Equation (1)) [16]:(1)$$M=\frac{I}{O}\;$$
where **I** = image size, and **O** = object size.

The image size is the final measured diameter, and the object size is the initial size of the iris (10.25 mm). Hence, the magnification of the iris was calculated by dividing the final measured diameter by the initial size of the iris.

### 2.5. Statistical Analysis

Analysis was performed using the SPSS Software 20 (SPSS Inc., Chicago, IL, USA). The descriptive statistics were calculated for the final measured diameter of the iris and magnification of the iris in various groups. A significant level at *p* value = 0.01 was chosen. One-way ANOVA, with post hoc, using the Sheffe test, was performed to compare the various sizes of the iris in different groups.

The relationship between the final measured diameter of the iris and magnification of the iris with the thickness of clear acrylic resin is calculated using OriginLab Software 10.5.110 (OriginLab, Northampton, MA, USA). License with the University of Central Florida.

## 3. Results

### 3.1. Final Measured Diameter of the Iris

Table 2 shows the descriptive statistics of the final measured diameter of the iris. It shows that, as the thickness of clear acrylic resin is increased, the final measured diameter of the iris is also increased. As the diameter is increased, the final measured diameter is increased in a greater magnitude. It showed that there was a significant difference in the final measured diameter of the iris among the groups (*p* value < 0.0001).

### 3.2. Magnification of the Iris in Ocular Prosthesis

Table 3 shows the descriptive statistics of the magnification of the iris in various groups. The magnifications of the iris for different groups were 1.065× for T1 (2 mm), 1.098× for T2 (4 mm), 1.161× for T3 (6 mm), 1.243× for T4 (8 mm), and 1.310× for T5 (10 mm).

The results of multiple comparisons showed that there was a significant difference in the magnifications of the iris among the groups (*p* value < 0.0001) (Table 4). The difference in iris sizes for different groups from the control were 0.67 mm for T1, 1.012 mm for T2, 1.652 mm for T3, 2.494 mm for T4, and 3.179 mm for T5. The magnifications of the iris for different groups were 1.065 for T1 (2 mm), 1.098 for T2 (4 mm), 1.161 for T3 (6 mm), 1.243 for T4 (8 mm), 1.310 for T5 (10 mm).

### 3.3. Relationship between the Final Measured Diameter and Magnification of Iris

The relationship between the final measured diameter of the iris and magnification of the iris with the thickness of clear acrylic resin is shown by Equations (2) and (3).
Final Diameter of Iris (mm) = (0.315 × Thickness of Clear Acrylic) **+** 10.189(2)
Magnification of Iris = (0.315 × Thickness of Clear Acrylic) **−** 0.060(3)

## 4. Discussion

Fabrication of the lifelike appearance of the ocular prosthesis is important for the patient’s esthetics and comfort [17]. New materials and better fitting ocular prostheses allow patients to wear prostheses with greater comfort and better appearance [17,18,19,20]. In this study, iris fabrication is done from black paper and attached directly over the ocular prosthesis. Various other iris fabrication methods are available in the literature, which include stitching method [7], printing [11,12], and iris button [21]. After adjusting the iris and sclera color, clear acrylic resin is added and cured.

The selected ocular size/dimension in our study is taken as a common ocular prosthesis size seen in clinical practice, as recommended by the oculoplastic surgeon. Too big an ocular prosthesis causes eye socket complications, and one that is too small size causes difficulty in ocular prosthesis fabrication and retention in the patient. In addition, selecting the proper size iris, in fabricating the ocular prosthesis, is one of the key steps in the fabrication of ocular prosthesis; however, due to magnification, often it results in an unesthetic ocular prosthesis and often readjustment or refabrication is needed. Hence, this study was done to provide information on the magnification of the iris by adding a specific amount of acrylic resin. Finally, we did a measurement of the iris in an ocular prosthesis with a spherical surface to simulate the exact real situation. The outer surface of the natural and prosthetic eyes are convex.

Although an ocular prosthesis can be fabricated from acrylic resin and ceramic, acrylic resin is the material of choice for the fabrication of ocular prostheses, due to easy handling, adaptation, good esthetics, ability to reline, and low cost [22,23]. Acrylic resin can simulate the natural color of the sclera, while the colorless resin is used to cover the characterization of blood vessels and artificial iris [23,24,25]. In addition, the acrylic resin has good physical and mechanical properties for the longevity of the ocular prosthesis [24,26,27,28]. The disadvantage of acrylic resin prosthesis is the material discoloration over time may be caused by intrinsic or extrinsic factors [29,30]. Hence, the patients need to change the prosthesis in about 5–7 years. The chemical and mechanical polishing of acrylic resin is done to reduce the ocular prosthesis surface roughness and accumulation of impurities and microorganisms [30].

In this research, it was found that, as the thickness of clear heat cure acrylic resin over the iris is increased on the ocular prosthesis, the magnification of the iris is also increased (Table 3). In addition, Equations (2) and (3) give the information on the final observed diameter of the iris by adding the amount of clear acrylic resin and magnification that occurred by adding the amount of clear acrylic resin, respectively. From this study, we found how much magnification can occur by adding 2, 4, 6, 8, and 10 mm of clear acrylic resin over the iris. Hence, these results can be used in the fabrication and relining of the ocular prostheses. This information and guideline on the magnification of the iris with the thickness of clear acrylic resin, provided in this research, will reduce the error in choosing the size of the iris.

Magnification of the iris is due to the refraction of light, with the clear acrylic resin acting as a convex lens. Magnification depends on both refractive index differences between the medium and curvature of the lens (Figure 5). The refractive index, or index of refraction (n), of clear acrylic is 1.48, which changes the direction of light by a change in speed. In the case of the ocular prosthesis, when the light from the actual iris passes the acrylic resin into the air, it bends and causes the visual perception of a larger iris.

Furthermore, this study provides the information of the final observed diameter of the iris and the magnification of the iris obtained through numerical calculations. Various factors that can affect the magnification of the iris in the ocular prosthesis are the thickness of clear acrylic resin, curvature of clear acrylic resin/shape of the ocular prosthesis, diameter of the iris, and refractive index differences between the clear acrylic resin and air (refractive index of PMMA/acrylic resin is 1.47 [31], with 1.0 for air) [32]. In the human eye, the refractive index of the cornea and overlying tear film are key factors affecting refraction and overall optical properties of the eye, and the normal refractive index of the human cornea is 1.376 (range from 1.335 to 1.4391) [33]. Similarly, the refractive index of PMMA is 1.47 [31]. Hence, the refraction of light in clear acrylic resin is comparable to the human cornea. If the ocular prosthesis is made from ceramic materials, the magnification results might have been slightly altered, due to the difference in the refractive index (refractive index of glass-ceramic is 1.55) [32].

In this study, the magnification was studied by taking the same diameter of the iris (10.25 mm) and same curvature of clear acrylic resin/ shape of the ocular prosthesis in all study groups, as well as the maximum 10 mm thickness of acrylic. Hence, future studies can be done by taking different sizes of iris and thicker acrylic resin in more samples, using different optical and geometrical parameters to study the magnification. In addition, it can be useful to do more specific characterization methods, and different optical and geometric parameters will be suitable for future investigation and characterizations. Additional methods are needed to study the swelling degree and thermo-sensitivity analysis of the acrylic resin.

## 5. Conclusions

The ocular prostheses, used for the prosthetic rehabilitation of ocular defects, are generally made from acrylic resin. As the thickness of clear acrylic over the iris is increased on the ocular prosthesis, the size of the iris is also increased, due to magnification through the clear resin. This research will provide guidelines on the magnification of the iris with the thickness of clear acrylic resin and help to reduce the error in the size of the iris, when fabricating and relining/adjustment of the ocular prosthesis. The results of this study can be also applied in magnification of any object in clear resin and is not limited to the iris in the ocular prosthesis.

## Figures and Tables

**Figure 1 jfb-13-00029-f001:**
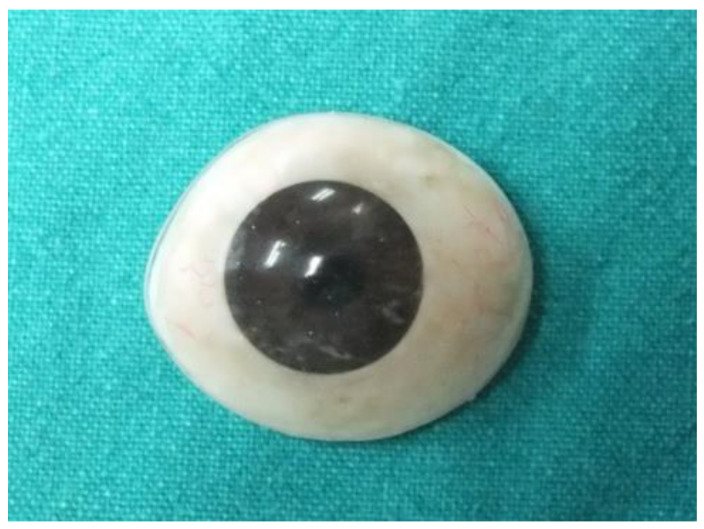
Ocular prosthesis.

**Figure 2 jfb-13-00029-f002:**
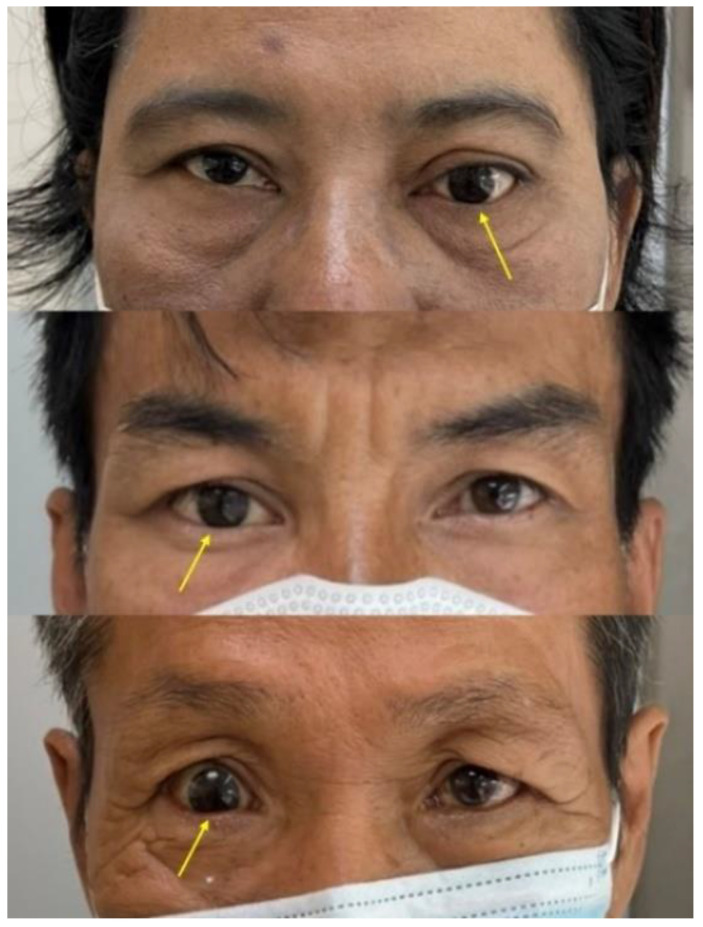
The iris size of the ocular prosthesis not matching the normal eye, due to the magnification of the iris from the clear acrylic resin.

**Figure 3 jfb-13-00029-f003:**
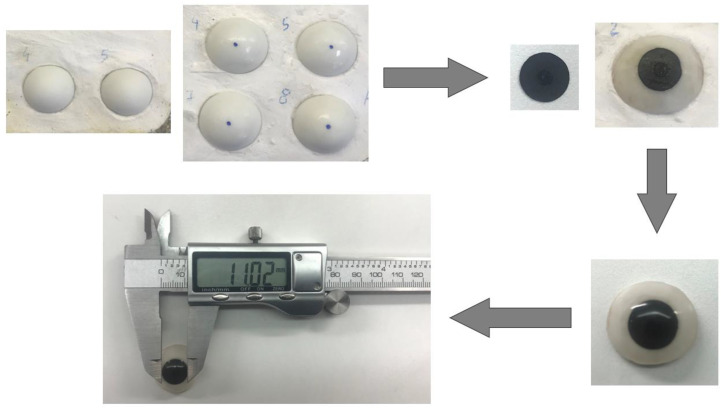
Fabrication of ocular prosthesis and measurement of the diameter of the iris using a digital vernier caliper.

**Figure 4 jfb-13-00029-f004:**
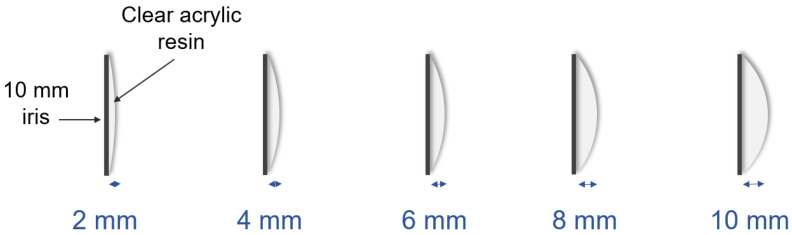
Thickness (2, 4, 6, 8, and 10 mm) of clear acrylic over the iris.

**Figure 5 jfb-13-00029-f005:**
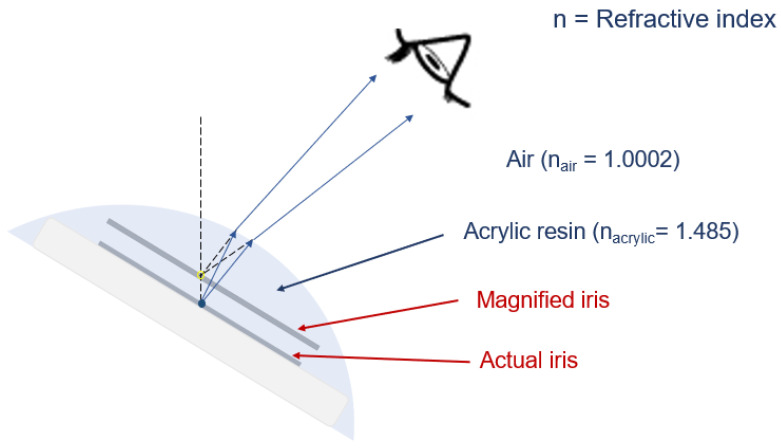
Magnification of the iris, due to the refraction of light, with the clear acrylic resin acting as a lens, due to the refraction of light.

**Table 1 jfb-13-00029-t001:** Study groups, according to the thickness of acrylic over the iris.

Groups (*n* = 60)	Acrylic Resin over the Iris
T0 (control)	No acrylic resin over the iris
T1	2 mm thickness clear acrylic resin over the iris
T2	4 mm thickness clear acrylic resin over the iris
T3	6 mm thickness clear acrylic resin over the iris
T4	8 mm thickness clear acrylic resin over the iris
T5	10 mm thickness clear acrylic resin over the iris

**Table 2 jfb-13-00029-t002:** Descriptive statistics of the final measured diameter of the iris in various groups.

Study Groups	Mean ± SD	SE	95% CI		Minimum	Maximum	*p* Value
			Lower Bound	Upper Bound			
T0	10.25 ± 0.04	0.01	10.22	10.28	10.18	10.33	<0.0001 *
T1	10.92 ± 0.05	0.08	10.88	10.96	10.82	11.00	
T2	11.26 ± 0.08	0.02	11.21	11.32	11.13	11.37	
T3	11.91 ± 0.09	0.03	11.84	11.97	11.79	12.07	
T4	12.75 ± 0.09	0.03	12.68	12.81	12.58	12.86	
T5	13.43 ± 0.06	0.02	13.38	13.48	13.32	13.53	

T0 = no acrylic above, T1 = 2 mm clear acrylic, T2 = 4 mm clear acrylic, T3 = 6 mm clear acrylic, T4 = 8 mm clear acrylic, and T5 = 10 mm clear acrylic. SD = standard deviation, SE = standard error, CI = confidence interval for mean. * Significant difference at *p* value < 0.01. Statistical analysis done using one-way ANOVA.

**Table 3 jfb-13-00029-t003:** Descriptive statistics of the magnification of the iris in various groups.

Study Groups (*n* = 60)	Magnification (Mean ± SD)	95% CI		Minimum	Maximum
		Lower Bound	Upper Bound		
No acrylic above	-	-	-	-	-
2 mm clear acrylic	1.06 ± 0.00	1.061	1.069	1.06	1.08
4 mm clear acrylic	1.01 ± 0.01	1.094	1.104	1.09	1.11
6 mm clear acrylic	1.16 ± 0.01	1.153	1.169	1.15	1.19
8 mm clear acrylic	1.24 ± 0.01	1.236	1.250	1.23	1.26
10 mm clear acrylic	1.31 ± 0.01	1.304	1.316	1.30	1.32

SD = standard deviation, CI = confidence interval for mean.

**Table 4 jfb-13-00029-t004:** The multiple comparisons of the magnifications of the iris among various groups.

Comparison Groups	Mean Difference	*p* Value
T0 vs. T1	1.06	<0.0001 *
T0 vs. T2	1.1	<0.0001 *
T0 vs. T3	1.160	<0.0001 *
T0 vs. T4	1.24	<0.0001 *
T0 vs. T5	1.31	<0.0001 *
T1 vs. T2	0.33	<0.0001 *
T1 vs. T3	0.96	<0.0001 *
T1 vs. T4	1.78	<0.0001 *
T1 vs. T5	0.24	<0.0001 *
T2 vs. T3	0.62	<0.0001 *
T2 vs. T4	0.14	<0.0001 *
T2 vs. T5	0.21	<0.0001 *
T3 vs. T4	0.08	<0.0001 *
T3 vs. T5	1.49	<0.0001 *
T4 vs. T5	0.06	<0.0001 *

T0 = no acrylic above), T1 = 2 mm clear acrylic), T2 = 4 mm clear acrylic, T3 = 6 mm clear acrylic, T4 = 8 mm clear acrylic, and T5 = 10 mm clear acrylic. * Significant difference at *p* value < 0.01. Statistical analysis done using one-way ANOVA, with post hoc, using the Sheffe test.

## Data Availability

The data presented in this study are available on request from the corresponding author.

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
