# Peer review of "Magnification of Iris through Clear Acrylic Resin in Ocular Prosthesis"

_jfb, 2022, doi:10.3390/jfb13010029_

Round 1

Reviewer 1 Report

The authors presented the investigation on the magnification of the iris with various thicknesses of clear heat cure acrylic resin. However, few points the authors should address before it can be accepted.

  • The author may extend fabrication methods for acrylic iris.
  • I am not clear on the method the authors used to study the magnification. Suggest to explain more on this measurement.
  • The authors may extend and give more rationale and interpretation for Table 2. Explain different between study groups.
  • How would the iris diameters affect the magnification?
  • Can the authors compare this study with others?

Author Response

Response to Reviewer 1 Comments

Point 1: The authors presented the investigation on the magnification of the iris with various thicknesses of clear heat cure acrylic resin. However, few points the authors should address before it can be accepted.

Thank you for your positive comments. All corrections in the Manuscript for Reviewer 1 are highlighted in Yellow color.

The author may extend fabrication methods for acrylic iris.

Response: Detail on iris fabrication of iris is added in the method. Line 103-106. More methods are discussed in the Discussion. Line 177-179.

Point 2: I am not clear on the method the authors used to study the magnification. Suggest to explain more on this measurement.

Response: The detail on the magnification is described in the method. Line 125-127.

Point 3: The authors may extend and give more rationale and interpretation for Table 2. Explain different between study groups.

Response: We thought Table 2 is not necessary, so it is removed and the results of the comparison among the group are added to the text. Line 142-144.

Point 4: How would the iris diameters affect the magnification?

Response: There is no study till now to give information on how the iris diameter affects the magnification. In this study, we only studied the magnification with a fixed diameter. It can be good to study in the future. We have added this in future studies. Line 239-241.

Point 5: Can the authors compare this study with others?

Response: There is no other study on the magnification of acrylic. So, we cannot compare.

Reviewer 2 Report

The article is interesting. It fits well with the subject of the journal. However, it needs improvement. The literature introduction is very short. I miss new trends in research on this type of materials. The purpose of the research was poorly emphasized. The cited bibliography could be more contemporary. Many works are more than 10 years old. In part of the discussion, I miss the reference to works on similar research topics. What is the scientific success of the authors of the article. Please complete this. 

Author Response

Response to Reviewer 2 Comments

The article is interesting. It fits well with the subject of the journal. However, it needs improvement. The literature introduction is very short. I miss new trends in research on this type of materials. The purpose of the research was poorly emphasized.

Corrections in the Manuscript are highlighted in Green color.

Response: The purpose of the research is improved.

The cited bibliography could be more contemporary. Many works are more than 10 years old. In part of the discussion, I miss the reference to works on similar research topics. What is the scientific success of the authors of the article. Please complete this. 

Response: As there is very little literature on this topic. We explored more and more latest literature are added in the discussion. The scientific success of the research is added to the discussion. Line 191-201, 188-190.

Reviewer 3 Report

Dear Authors,

The manuscript ID 46-jfb-1606379, entitled "Magnification of iris through clear acrylic resin in ocular prosthesis was interesting and useful study. After careful reading, in my opinion, this manuscript needs major improvements according specific comments, as followed:

The Abstract provides an important overview of the manuscript, but is necessary to be shorter (about 200 words maximum) and revised with one sentence with background, after that should be highlighted main goals, applied methods, part of results and meaningful conclusion. Process of manufacture in detail is redundant (lines 25-29) and it is needed to be shorter.

In the Introduction part, authors have presented a short state of the art, main goals of this study, but some of expected results should be added.

Materials and Methods part provides necessary data about methods for fabrication of ocular prostheses and measurement of the final diameter of the iris in ocular prostheses, calculation of the magnification and applied statistical analysis. Authors should insert data about applied materials and instruments before subsection 2.1.

Results part, presented in 4 tables, provides data about final measured diameter of the iris, magnification of the iris in ocular prosthesis and relationship between the final measured diameter and magnification with the thickness of acrylic resin, defined with 2 equations.

  • Please, it is needed to revise the following sentence (line 90): "At first, dental molds were prepared with dental stones."
  • Lines 160-161: It is needed to divide subsection title from the main text.
  • Lines 164-165: Why is the number (0.315) in Equations 2 and 3 given in parentheses? Please, mathematical operations are needed to be applied in Equations.

Discussion part provides analyses and comments about the obtained results.

  • First part of discussion (lines 167-174) should be moved in the part 1. Introduction.
  • Line 177: It is needed to correct numbers of Equations as 2 and 3, instead of 3 and 4.
  • The authors should give an opinion on which combination of obtained results would be the most suitable for application.
  • Lines 204-205: It would be useful to add more specific characterization methods and which different optical and geometric parameters will be suitable for future investigation and characterizations.
  • Did the authors consider the fact that methacrylate-based materials swell over time and to examine behavior/potential volume increase over time. For this statement, additional methods are needed for confirmation, e.g. the swelling degree and thermo-sensitivity analysis.

In the Conclusion part the article's main findings and interpretations of analyzed data were summarized. The sentence (lines 191-192): "The results of this study can be also applied in magnification of any object in clear resin not limited to the iris in the ocular prosthesis" should be moved in Conclusion.

Authors cited 21 references, and one of them is relevant self-citation. It is needed to add the digital object identifier (DOI) for all references where available.

  • It is needed to avoid 1st person plural and rewrite sentences in 3rd person plural and passive voice (lines 21, 175 and 199).

After revision by the authors, this manuscript can be considered for publishing in the "Journal of Functional Biomaterials".

Best regards!

Reviewer

Author Response

Response to Reviewer 3 Comments

The manuscript ID 46-jfb-1606379, entitled "Magnification of iris through clear acrylic resin in ocular prosthesis” was interesting and useful study. After careful reading, in my opinion, this manuscript needs major improvements according specific comments, as followed:

Response: Thank you for your positive comments. This manuscript was corrected as per comments from the other 3 reviewers. Your (Reviewer 3) comments are really important and valuable and the corrections in this Manuscript are highlighted in Yellow color.

The Abstract provides an important overview of the manuscript, but is necessary to be shorter (about 200 words maximum) and revised with one sentence with background, after that should be highlighted main goals, applied methods, part of results and meaningful conclusion. Process of manufacture in detail is redundant (lines 25-29) and it is needed to be shorter.

Response: The abstract is edited and reduced the word count to 248. Reducing more then this would not make clear abstract as we had to include background, aim, method, results, and conclusion.

In the Introduction part, authors have presented a short state of the art, main goals of this study, but some of expected results should be added.

Response: Expected results are added in the introduction. Line 70-74.

Materials and Methods part provides necessary data about methods for fabrication of ocular prostheses and measurement of the final diameter of the iris in ocular prostheses, calculation of the magnification and applied statistical analysis. Authors should insert data about applied materials and instruments before subsection 2.1.

Response: The method section has been edited and improved. The data about the applied materials and instruments are added (Subsection 2.1)

Results part, presented in 4 tables, provides data about final measured diameter of the iris, magnification of the iris in ocular prosthesis and relationship between the final measured diameter and magnification with the thickness of acrylic resin, defined with 2 equations.

  • Please, it is needed to revise the following sentence (line 90): "At first, dental molds were prepared with dental stones."
  • Response: This line is revised. Line 98.
  • Lines 160-161: It is needed to divide subsection title from the main text.
  • Response: Line 165-167.
  • Lines 164-165: Why is the number (0.315) in Equations 2 and 3 given in parentheses? Please, mathematical operations are needed to be applied in Equations.
  • Response: Parentheses are removed “(0.315)” and “x” is added to show multiplication.

Discussion part provides analyses and comments about the obtained results.

  • First part of discussion (lines 167-174) should be moved in the part 1. Introduction.
  • Response: Detail on iris fabrication of iris is added in the method. Line 103-106. More methods are discussed in the Discussion. Line 177-179.
  • Line 177: It is needed to correct numbers of Equations as 2 and 3, instead of 3 and 4.
  • Response: Numbering is corrected.
  • The authors should give an opinion on which combination of obtained results would be the most suitable for application.
  • Response: We have added these for application in the Discussion. Line 200-209.
  • Lines 204-205: It would be useful to add more specific characterization methods and which different optical and geometric parameters will be suitable for future investigation and characterizations.
  • Response: Added in the end of the Discussion. Line 241-243.
  • Did the authors consider the fact that methacrylate-based materials swell over time and to examine behavior/potential volume increase over time. For this statement, additional methods are needed for confirmation, e.g. the swelling degree and thermo-sensitivity analysis.
  • Response: Added in the end of the Discussion. Line 243-245.

In the Conclusion part the article's main findings and interpretations of analyzed data were summarized. The sentence (lines 191-192): "The results of this study can be also applied in magnification of any object in clear resin not limited to the iris in the ocular prosthesis" should be moved in Conclusion.

Response: This line is moved to the Conclusion. Line 250-252.

Authors cited 21 references, and one of them is relevant self-citation. It is needed to add the digital object identifier (DOI) for all references where available.

  • Response: Fore referencing we followed the manuscript guideline. Use used MDPI endnote style in referencing.
  • It is needed to avoid 1st person plural and rewrite sentences in 3rd person plural and passive voice (lines 21, 175 and 199).
  • Response: Line 22-23, 201, 232-233.

After revision by the authors, this manuscript can be considered for publishing in the "Journal of Functional Biomaterials".

  • Response: Thank you.

Reviewer 4 Report

This work studies the various types of ocular prostheses made with acrylic resin for the magnification of the iris. Statistical modeling is employed in the work to analyze the results. This work is within the scope of the journal.

The language is overall good, with few changes required throughout the manuscript.

The references in the study need to be enriched. This will help highlight the contribution to the field of this work, which is missing and should be analyzed and presented in the introduction section of the manuscript. Apart from the novelty of the work, the purpose of conducting such research should be more clearly presented.

In the last paragraph of the introduction section what is done, how, and what was found should be presented.

The specs of the resin used should be described and explained why this is suitable for such applications.

Why dental molds were used?

Why these dimensions mentioned are suitable for the work? What is the difference between them regarding the result in the eye?

Is the measurement with the caliper a suitable method for the work since a spherical surface is involved?

Equation 1 should be further explained, and a reference should be added.

The parameters affecting the magnification of the iris should be further discussed and evaluated.

How the selection of the specific material affects the obtained results? What is the expected variation of the results if some other material was used?

Apart from the measurements on the geometry, have these been tested in vivo, and what were the results?

Author Response

Response to Reviewer 3 Comments

Point 1: This work studies the various types of ocular prostheses made with acrylic resin for the magnification of the iris. Statistical modeling is employed in the work to analyze the results. This work is within the scope of the journal. The language is overall good, with few changes required throughout the manuscript.

Response: Thank you. All corrections in the Manuscript are highlighted in Turquoise.

Point 2: The references in the study need to be enriched. This will help highlight the contribution to the field of this work, which is missing and should be analyzed and presented in the introduction section of the manuscript. Apart from the novelty of the work, the purpose of conducting such research should be more clearly presented.

Response: As no similar works have been done in the past, we cannot find many references. The references are added wherever we can add. Ref 16. The purpose of this work is added and highlighted.

Point 3: In the last paragraph of the introduction section what is done, how, and what was found should be presented.

Response: In the last paragraph of the introduction, a brief on what, how, and what was found is added.

Point 4: The specs of the resin used should be described and explained why this is suitable for such applications.

Response: The details of the resin which makes it suitable for such application are added. Line 191-201.

Point 5: Why dental molds were used?

Response: The “dental molds” were changed to “ocular molds”. Line 97.

Point 6: Why these dimensions mentioned are suitable for the work? What is the difference between them regarding the result in the eye?

Response: The selected ocular size/dimensions are taken as a common ocular prosthesis size used in clinical practice as recommended by the oculoplastic surgeon. Too big ocular prosthesis causes eye socket complications and too small size causes difficulty in ocular prosthesis fabrication and retention in the patient. Added in the Discussion. Line 180-183.

Point 7: Is the measurement with the caliper a suitable method for the work since a spherical surface is involved?

Response: We did the measurement of the iris in an ocular prosthesis with the spherical surface to simulate the exact real situation. The outer surface of the natural eye and the prosthetic eye is convex. Hence, we measured on the convex surface. We added in the Discussion. Line 188-190.

Point 8: Equation 1 should be further explained, and a reference should be added.

Response: Equation 1 is explained. This equation is obtained from our results, so there is no reference.

Point 9: The parameters affecting the magnification of the iris should be further discussed and evaluated.

Response: The parameters affecting the magnification of the iris are discussed in the Discussion. Line 226-236.

Point 10: How the selection of the specific material affects the obtained results? What is the expected variation of the results if some other material was used?

Response: At present, generally acrylic resin is used for the fabrication of ocular prostheses. If the ocular prosthesis is made from ceramic materials, the magnification results might have been slightly altered due to the difference in the refractive index (the refractive index of glass-ceramic is 1.55). Added in the Discussion. Line 234-236.

Point 11: Apart from the measurements on the geometry, have these been tested in vivo, and what were the results?

Response: If the results were tested in vivo, the results would be the same, as the ocular prosthesis is made in the lab and inserted in the patients. So, the same results are expected.

Round 2

Reviewer 1 Report

The authors provide sufficient responses to the queries. 

Author Response

Thank you very much.

Reviewer 2 Report

The article can be accepted in its present form.

Author Response

Thank you very much.

Reviewer 3 Report

Authors team in revised manuscript ID jfb-1606379, entitled "Magnification of iris through clear acrylic resin in ocular prosthesisis improved required data and provided sufficient responses according previous suggestions. After careful reading, in my opinion, this manuscript needs minor improvements according specific comment, as followed:

  • Lines 168-169 and 202: It is needed to correct numbers of Equations as 2 and 3, instead of 1 and 2.

This manuscript  can be published in journal "Journal of Functional Biomaterials".  

Best regards!

Reviewer

Author Response

Authors team in revised manuscript ID jfb-1606379, entitled "Magnification of iris through clear acrylic resin in ocular prosthesis” is improved required data and provided sufficient responses according previous suggestions. After careful reading, in my opinion, this manuscript needs minor improvements according specific comment, as followed:

  • Lines 168-169 and 202: It is needed to correct numbers of Equations as 2 and 3, instead of 1 and 2.

Thank you for your positive comments. Your (Reviewer 3) comments are really important and valuable and the corrections in this Manuscript are highlighted in Yellow color.

  • The Equations numberings are corrected. Line 168-169, 202.

Reviewer 4 Report

The revised version of the manuscript is significantly improved in its technical aspects. All the comments of this reviewer have been adequately replied and corresponding amendments have been made in the revised version of the manuscript. So, manuscript can be published in its current form.

Author Response

Thank you very much.